# Histological and Molecular Evidence of the Positive Performance of Glycerol-Plasticized Chitosan-Alginate Membranes on Skin Lesions of Hyperglycemic Mice

**DOI:** 10.3390/polym14214754

**Published:** 2022-11-06

**Authors:** Flavia Figueiredo Azevedo, Thiago Anselmo Cantarutti, Paula de Freitas Rosa Remiro, Beatriz Barbieri, Rafael Abboud Azoubel, Mariana Harue Taniguchi Nagahara, Ângela Maria Moraes, Maria Helena Melo Lima

**Affiliations:** 1School of Nursing, University of Campinas, Campinas 13083-887, SP, Brazil; 2Department of Engineering of Materials and of Bioprocess, School of Chemical Engineering, University of Campinas, Campinas 13083-852, SP, Brazil

**Keywords:** chitosan, alginate, wound healing, brown adipose tissue, Diabetes Mellitus

## Abstract

The purpose of this study was to investigate tissue repair of excisional wounds in hyperglycemic animals treated with chitosan-alginate membranes (CAM) produced in the presence of glycerol. 8-week C57B1 male mice were divided into normoglycemic animals with a 0.9% saline solution topical treatment (CTSF); hyperglycemic animals with 0.9% saline solution topical treatment (DMSF) and hyperglycemic animals with glycerol-plasticized chitosan-alginate membrane topical treatment (DMCAM). On post-wound day three, the DMCAM group presented a lower number of leukocytes, mature mastocytes, a higher number of vessels (*p* < 0.05), and active mastocytes (*p* < 0.05) when compared to the CTSF and DMSF groups. There were no differences regarding the distribution, deposition, organization, and thickness of collagen fibers. On day 7 there were no differences in the analysis of fibroblasts, mastocytes, and TGF−β1 and VEGF expressions among the groups. Regarding collagen fibers, the DMCAM group presented slight red-orange birefringence when compared to the CTSF and DMSF groups. On day 14 there was a slight concentration of thinner elastic fibers for the DMCAM group, with a greater reorganization of papillary skin and improved red-orange birefringence collagen fibers, as well as net-shaped orientation, similar to intact skin. In addition, improved elastic fiber organization distributed in the entire neo-dermis and a larger presence of elaunin fibers were observed, in a similar pattern found in the intact skin. The use of CAM in cutaneous lesions boosted tissue repair since there was a smaller number of inflammatory cells and mastocytes, and an improvement in collagen deposition and collagen fibers. These results demonstrate the high potential of plasticized chitosan-alginate membrane for skin wound dressing of hyperglycemic patients.

## 1. Introduction

Diabetes Mellitus (DM) is a highly prevalent metabolic disorder worldwide. It is defined as persistent hyperglycemia due to a deficiency in insulin production or action. When not adequately treated, it can lead to severe complications, such as inhibiting the healing process [1].

People with DM present slow healing and in some cases their lesions may never heal [2]. Difficulty in healing is related to several factors, mainly due to failure during the inflammatory process. In addition, the concomitant occurrence of peripheral arterial disease, hypoxia, and peripheral neuropathy contribute to delays in healing [3]. Besides that, several mechanisms have been identified as critical factors in the delay of the healing process in people with diabetes, such as changes in the response of angiogenesis and reactive oxygen species (ROS), a decrease in the production and availability of Nitric Oxide (NO) [4], lower expression of the insulin receptor and proteins involved in the insulin signaling cascade in the wounded tissue [5].

Treatment of skin lesions has evolved in the last three decades after the introduction of the first dressing specially designed for hyperglycemic patients. Today, there are a variety of dressings available, along with advanced technologies that contribute to wound healing [6,7]. Studies show that dressings with biomaterials designed for this purpose contribute to the healing process by offering protection against colonization/infection, providing a moist environment, avoiding trauma, as well as releasing bioactive substances [8,9,10,11]. In particular, membranes consisting of chitosan and alginate can modulate wound healing, stimulating collagen deposition and epithelialization [12]. In a study with normoglycemic animals, it was observed that a membrane with chitosan and alginate was able to modulate the inflammatory phase, fibroplasia, and collagenases, which contributed to the healing process [12].

Chitosan is a polysaccharide obtained from the deacetylation of chitin, which can be found in the exoskeleton of crustaceans [10,13,14], as well as in the cell walls of some fungi. It is biologically renewable, biodegradable, and non-antigenic. It can modulate the wound healing process by itself, improving inflammatory cells, macrophage, and fibroblast functions [15,16,17]. The mechanical resistance of films made only of chitosan is limited, as well as its capacity to absorb bodily fluids. Thus, an alternative to avoid these and even other limitations is combining chitosan with other compounds, such as alginate [10,11,12].

Alginate, a hydrophilic polymer that can present anionic characteristics in pH conditions that are superior to approximately 4.0, can be extracted from brown seaweed and is biocompatible and biodegradable under normal physiological conditions, as well as able to maintain a physiologically moist microenvironment that can boost granular tissue formation [15,16,18]. Alginate dressings have been a part of clinical practice, since they act on the ionic exchange between calcium ions commonly found in alginate matrices and sodium exudate ions, absorbing excess liquid and forming a gel [19].

When combined in the aqueous phase, chitosan and alginate form complex polyelectrolytes (PEC) that can alter the tendency of the wound to swell. An improvement in structural resistance and mechanical stability can be observed [12]. Interest in the use of chitosan and alginate PEC has increased in the field of wound dressings and in the formulation of soft tissues that function as structures for the migration, proliferation, and organization of cells in the microenvironment of the wound [19,20].

In a study with hyperglycemic animals, it was observed that the chitosan-alginate membrane modulated the pro-inflammatory IL-1α, IL-1β and G-CSF cytokines (an activating factor in the production of granulocyte colonies) on the 1st and 2nd days after topical treatment of skin wounds [21]. On day 5, the same cytokines presented a decrease in concentration when compared to the control group. On the other hand, there was an increase in VEGF and α–SMA expression (smooth muscle alpha-actin) [21].

It was observed that the chitosan-alginate membranes used previously [21], when dry, had limited flexibility, making it difficult to handle and place the dressing on the surface of the wound. Hence, in this study, to make the application easier, glycerol was included in the biomaterial formulation, since it is a biocompatible plasticizer. Therefore, the purpose of this study was to investigate tissue repair of excisional wounds in hyperglycemic animals treated with chitosan-alginate membranes produced in the presence of glycerol. The search for new therapies is needed for the treatment of wounds that do not heal easily, as well as to increase understanding of cell mechanisms to advance pre-clinical studies for clinical practices. The use of flexible dressings, of easy application, comfortable for the patient and effective in terms of improving healing, as the one discussed herein, is of foremost importance and we hope to show in the present work the advantages of this formulation.

## 2. Materials and Methods

This work consists of an experimental study with the scientific use of animals in which an intervention was carried out. A control group was also used. The study followed recommendations of the ARRIVE 2.0 guide from the National Center for the Replacement, Refinement, and Reduction of Animals in Research (NC3Rs) [22].

### 2.1. Sample Size, Randomization, and Blinding

Ninety animals were obtained from the Multidisciplinary Center for Biological Investigation in the Field of Laboratory Animal Sciences at the University of Campinas (CEMIB/UNICAMP), Campinas, SP, Brazil. The mice were weighed and separated according to the probabilistic method of random choices into three groups of 30 animals each. For the application of blind criteria and bias reduction in the study, analyses of results were carried out by two researchers who did not know the groups.

### 2.2. Group Composition and Diabetes Induction

C57BL/6/JUinb male mice, between 8 and 10 weeks old and an approximate weight of 25 g, were kept in standard lighting conditions, in light and darkness cycles, at a temperature of 22 ± 2 °C and with access to water and food ad libitum. Ten animals were allocated to each group according to the description below:Normoglycemic animals submitted to topical treatment of 0.9% saline solution (CTSF);Hyperglycemic animals submitted to topical treatment of 0.9% saline solution (DMSF);Hyperglycemic animals submitted to topical treatment of plasticized chitosan-alginate membranes (DMCAM).

The animals allocated to the DMSF and DMCAM groups were submitted to diabetes induction with the administration of 50 mg/kg of streptozotocin (STZ) (Sigma^®^-ST Luis/MO, USA), given intraperitoneally (IP) after reconstitution using 0.1 M citrate buffer (pH 4.5), during five consecutive days (5 × 50 mg/kg) [23]. The CTSF animals were submitted to intraperitoneal administration (IP) of 0.1 M citrate buffer (pH 4.5) at the same dosage described above, for five consecutive days.

On the first day of diabetes induction, all animals were allocated into individual boxes for detailed observation of each animal’s health condition. After day 5 of STZ administration, the animals remained under observation for 14 days, with access to water and food ad libitum. After this period, the animals in the three groups fasted for 6 h and their glycemic level was analyzed with a glucose meter (Accu-chek Active) using tail blood samples. Animals presenting capillary glycemia ≥ 250 mg/dL were considered hyperglycemic [23].

### 2.3. Criteria for Inclusion and Exclusion

Animals submitted to 1M citrate buffer (pH 4.5) via IP, with capillary glycemia ≤ 120 mg/dL and allocated to the CTSF group were included in the study. Animals that were given STZ via IP with capillary glycemia ≥ 250 mg/dL were allocated to the DMSF and DMCAM groups.

Animals with signs of infection in the wound bed, with purulent exudates, edemas, and perilesional hyperemia were excluded. DMCAM group animals in which there was chitosan-alginate membrane loss during any phase of the healing process were also excluded. The animals classified in the exclusion criteria were euthanized with lethal doses of intraperitoneal (IP) Ketamine Hydrochloride (Ketalar, Parke-Davis, Detroit/MI, USA) and Xylazine Hydrochloride (União Química Farmacêutica Nacional S/A, São Paulo/SP, Brazil).

### 2.4. Producing the Excisional Wound

After confirming the glycemic state, the excisional wound was produced. Animals were anesthetized with 180 mg/kg of Ketamine Hydrochloride (Ketalar, Parke-Davis, Detroit/MI, USA) and 8 mg/kg of Xylazine Hydrochloride (União Química Farmacêutica Nacional S/A, São Paulo/SP, Brazil), via IP. After confirming anesthetic induction through the absence of corneal reflexes, a trichotomy of the interscapular region was performed. Skin antisepsis was done with a degerming solution at 2% and skin demarcation using a mold of 1 cm in diameter. The skin was then surgically removed immediately.

### 2.5. Pain Assessment

Right after the surgical procedure was performed to produce the excisional wounds, the animals in groups CTSF, DMSF, and DMCAM were given Tramadol hydrochloride (20 mg/kg) [24], via IP, every 24 h for three consecutive days. Application of the drug therapy with Tramadol hydrochloride was based on the protocol for pain assessment in mice and consisted in analyzing expressions called “Units of Action.” Thus, during this period, the following signs of pain were analyzed individually in each animal: eye closure, bulging nose and cheeks, and ear position [25].

### 2.6. Treatment

Regardless of the group, all wounds on animals were submitted to the topical application of 0.9% saline solution (SS 0.9%) daily. Wound occlusions with dressings or secondary bandages were not done. Animals allocated to the DMCAM group were submitted to the application of chitosan-alginate membrane previously hydrated with 0.9% SS immediately after creating the excisional wound, only once.

### 2.7. Production and Characterization of Plasticized Chitosan and Alginate Membrane

The membrane was prepared using a 1:1 mass ratio of chitosan (C) (medium molecular weight, lot number STBF3507V, Sigma-Aldrich, ST Luis/MO, USA) and alginate (A) (medium viscosity, lot number SLBV9536, Sigma-Aldrich, ST Luis/MO, USA). Initially, a solution of alginate at 1% (*m*/*v*) in deionized water was prepared. In parallel, a solution of chitosan, also at 1% (*m*/*v*), dissolved in aqueous acetic acid solution at 1% (*v/v*), was prepared and filtered with the aid of a 50 mL glass Buchner funnel with a class 2 porous plate. An aliquot of 110 mL of the alginate solution was transferred to a tank and 110 mL of the chitosan solution was slowly added to the vessel with a peristaltic pump (Minipuls 3, Gilson, Middleton, USA) at a flow rate of 6 mL/min, under a mixing rate of 500 rpm (propeller stirrer Q250M2, Quimis, Diadema/SP, Brazil), at room temperature (around 25 °C). The tank contents were further mixed for 30 min, and afterward, the pH of the mixture was corrected to 7 with the slow addition of a 2 mol/L NaOH aqueous solution by dripping. After neutralization, 3 mL of glycerol were added.

To promote ionic crosslinking, 15 mL of 0.5% (*wt/v*) CaCl_2_ was slowly dripped in the mixture under mixing at 500 rpm. Then, the mixing rate was increased to 1000 rpm for 10 min. After this procedure, the solution was deaerated for 12 h (vacuum pump TE-058, Tecnal) to ensure the removal of trapped air bubbles, aliquoted in 100 g samples in two Petri dishes, and dried in an oven with air circulation (410/3NDE, Nova Ética) for 24 h at 37 °C. Both membranes went through a secondary crosslinking process with 1% CaCl_2_ (*m*/*v*) for 30 min and then, washed with deionized water (Milli-Q Academic, Millipore, Burlington/MA, USA) for an additional 30 min period and dried at room temperature.

The final membranes were then characterized regarding their aspect by visual inspection and by a scanning electronic microscope coupled to an energy dispersion X-ray detector (SEM model Quattro S from Thermo Fisher Scientific at 15 kV tension, 31 pA current, and EDS model ANAX-60P-B from Thermo Scientific UltraDry, Waltham/MA, USA), after coating with a 200 Å layer of gold (Sputter Coater model K450 EMITECH). Membrane thickness in dry and wet states (after exposure to saline at 0.9% at 37 °C for 24 h) was determined by measuring with a digital micrometer (Mitutoyo), and swelling degree and mass loss by gravimetry (Analytical Plus balance, Ohaus, Parsippany/NJ, USA) after exposure to 0.9% saline solution for 24 h and 7 days, respectively. The membranes were also analyzed regarding rugosity (rugosimeter SJ-201, Mitutoyo, Suzano/SP, Brazil) on both sides (faces exposed to the air and to the Petri dish while drying) and contact angle with water using a CAM-Micro Angle Meter (TANTEC). The mechanical properties of the membranes were determined using 8 × 1 cm^2^ samples in a texturometer (TA.XTPlus, Stable Micrometrics, Surrey, UK), with a load cell of 50 kg, a distance between grips of 5 cm and a test speed of 1 mm/s. Membrane opacity was analyzed using a colorimeter (ColorQuest XE, HunterLab, Virgínia, USA).

### 2.8. Assessment of Wound Closure

To assess the healing process, the wounds were photographed by the same evaluator immediately after wound preparation (day 0) and on post-wound days 3, 7, and 14, with a Canon Power Shot^®^ camera (model SX400 IS 16MP × Optical zoom).

Images were digitized and the wound area was measured using the ImageJ 1.49v software (National Institutes of Health, Bethesda, MD, USA). Results of the original wound area were expressed in square centimeters (cm^2^). Analyses of the wound area were carried out by a researcher with no knowledge of the groups for blinding at this stage of the study.

### 2.9. Tissue Extraction

On days 3, 7, and 14, post-wound, animals were once again submitted to the anesthetic procedure, and surgical extraction of the skin in the wound area was carried out, preserving the intact skin in the perilesional region. Skin extraction of the wound was done in experiment 1 on post-wound day 3; in experiment 2, on day 7; and in experiment 3, on post-wound day 14 [21].

At the end of each experiment, animals were euthanized with an overdose of the anesthetic.

### 2.10. Histological Analyses

Fragments of the wounded tissues were laid out on cork right after extraction and attached with pins so as not to alter tissue morphology. After that, they were placed in individual vials in a 3.7% formaldehyde solution (pH 7.2) during 8 h for tissue fixation. Afterward, the skin fragments were stored in individual cassettes and kept in a 70% alcohol solution overnight. Subsequently, the material was submitted to a sequence of 95% alcohol, 100% alcohol, and xylene washes. After the washes, the material was submitted to a paraffin wash and inclusion at 60 °C. Cuts with 5 μm thickness were made with a microtome. After laying out the material on microscope slides, staining was done with Hematoxylin and Eosin (H&E), Alcian Blue + Safranine, Picrosirius red, and Resorcin- Fuchsin and Weigert.

Histopathological analysis was performed according to a scoring system using, respectively, the signs −, +, ++, +++ respectively for absent, light, moderate and intense for the presence of leukocytes, fibroblastss, blood vessels, mature mastocytes, and immature mastocytes.

### 2.11. Western Blotting Analysis

To carry out Western Blotting analysis, the tissue taken from the wounds during the extraction periods described previously was homogenized on a pH 7.4 buffer containing 1% of Triton X-100, 100 mmol/L of Tris, 100 mmol/L of sodium pyrophosphate, 100 mmol/L of sodium fluoride, 10 mmol/L of EDTA, 10 mmol/L of sodium orthovanadate, 2 mmol/L of PMSF and 0.01 mg/mL aprotinin using a Polytron PT A20S homogenizer (Brinkmann Instruments, Riverview, FL, USA).

Tissue extracts were centrifuged at 11,000 rpm, at 4 °C for 40 min for the removal of the insoluble material. After these procedures, the supernatant was aspirated and its protein content was quantified with bicinchoninic acid reagent (BCA assay). The tissue extracts were mixed with Laemmli buffer plus 200 mmol/L of dithiothreitol (DTT) in a 4:1 proportion (*v*/*v*) and were submitted to electrophoresis in a polyacrylamide gel in the presence of sodium dodecyl sulfate (SDS—PAGE 8%, 12%, and 15%) in a minigel device. Standard molar mass markers were used (Precision plus protein standards—Bio-RAD dual color 1610374) to estimate the molar weight of the components.

The transference of proteins separated in the gel to a nitrocellulose membrane was done with a BioRad equipment for 2 h at 120 V in an ice wash. 0.1% SDS was added to the buffer to improve the dissolution of high molar mass proteins. Unspecific connections of proteins to the nitrocellulose membrane were minimized with incubation in a blocking solution (5% of Molico^®^ (NESTLE, Vevey, Switzerland) skim milk, 10 mmol/LM of Tris, 150 mmol/L of NaCl, and 0.02% of Tween 20) at room temperature for 2 h, under mild shaking.

The nitrocellulose membranes were incubated with the following specific antibodies: VEGF (Ab1316; 1:250) and TGF-beta 1 (Ab92486; 1:1000) diluted in a blocking solution (3% BSA), overnight, at a temperature of 4 °C, under mild shaking, and subsequently washed with the same BSA solution for 30 min. After washing, the membranes were incubated with antibodies conjugated with peroxidase for two h at room temperature, followed by a chemiluminescence detection solution as described in the commercial kit protocol. Light emission was analyzed using the MCHEMIBIS detection system (Bio-Imaging Systems Ltd., Jerusalem, Israel) and the band intensity on revelation was determined through band optical densitometry analysis using the GelAnalyzer2010a program. Results were normalized by the Beta-actin (15G5A11/E2) primary antibody.

Analyses of the band intensity of each nitrocellulose membrane incubated with the antibodies described above, in accordance with the blind strategies planned for the study, were carried out by researchers who did not know the treatment groups.

### 2.12. Data Analysis

The results are presented as the mean ± S.E.M. The normality of the data was confirmed using the Shapiro-Wilk test. Parametric data were compared using two-way ANOVA or one-way ANOVA, followed by a Tukey multiple comparison test. Statistical analyses were conducted with GraphPad Prism software version 6.01 (GraphPad Software 6.01, Inc., San Diego, CA, USA). Data with *p*-values of below 0.05 were considered significantly different.

### 2.13. Ethical Aspects

This study was approved by the Ethics Committee for Animal Use at the University of Campinas (CEUA/UNICAMP) No. 4935-1/2018, in accordance with Law No. 11794/October 8, 2008, which establishes procedures for the scientific use of animals; and with Decree No. 6899/July 15, 2009, as well as norms published by the National Council for the Control of Animal Experiments (CONCEA).

## 3. Results

### 3.1. Membrane Characteristics

The visual aspect of the membrane is shown in Figure 1, along with its microstructure characteristics, and its properties are summarized in Table 1.

We can observe that the membrane is homogeneous, transparent (null opacity), thin (thickness considerably below 1 mm even when wet, and because of that, potentially comfortable when applied). The membrane has a relatively smooth surface, showing low rugosity values on both sides, which may also contribute to the comfort of the dressing. It shows a lamellar microstructure and it is mechanically strong (tensile strength above 80 MPa) and flexible (Young’s modulus of 4024 MPa) (Figure 1). In addition, the membrane is also highly hydrophilic, with a water contact angle significantly below 90°, and has adequate capacity to absorb saline solution, being sufficiently stable in it for up to a week.

The membrane used herein, to which glycerol was added as a plastifier, showed many characteristics similar to those of the dressing free of glycerol used in previous work [21], such as thickness, transparency, and elongation at break, but was significantly less rigid, as expected, being of easier application and conformation to the lesion site. The membrane described in the present work was also much more resistant to tensile strength (almost three times more) and capable of absorbing a little less than half of the saline solution absorbed by the dressing described previously (which reached 14 ± 1 g/g) [21].

### 3.2. Analysis of the Effect of Plasticized Chitosan-Alginate Membranes on the Kinetics of Skin Wound Healing

The initial area of the excisional wound (day 0) was 1 cm^2^. On day 3, a significant difference in terms of wound area was observed in the CTSF group in comparison to the DMSF group (*p* < 0.05). There was no statistical difference in the lesioned area of the DMCAM group when compared to the remaining groups. On day 7 there were no significant differences among the groups. On post-wound day 14, the DMCAM group presented a scar with regular contours, which looked aesthetically better when compared to the CTSF and DMSF groups (Figure 2). In the macroscopic evaluation of the first post-wound days, it was observed that in the DMCAM group the lesions presented smaller edema, hyperemia, and more regular edges when compared to groups CTSF and DMSF. On day 14, the scar in the DMCAM group was discreetly smaller in relation to CTSF and DMSF, though there was no statistical difference (Figure 2).

### 3.3. Qualitative and Morphometric Analyses of Skin Wounds Submitted to Topical Treatment with Plasticized Chitosan-Alginate Membranes

For the qualitative analysis of the skin wounds after days 3, 7, and 14, H&E and Alcian Blue + Safranin staining were used for the assessment of the general aspect of the tissue, such as reepithelization, inflammatory infiltrate, mature and immature mastocytes, as well as the number of vessels in the wound area.

Qualitative and quantitative histopathological analysis of the skin collected on post-wound day 3 and stained with H&E showed that the animals in the DMCAM group presented a smaller number of leukocytes and mature mastocytes, as well as an increased number of vessels when compared to those in groups CTSF and DMSF (*p* < 0.05) (Figure 3A,B). In addition, a higher number of active mastocytes (*p* < 0.05) was observed in the group DMCAM when compared to the other groups. No statistical differences were noticed in the expression of TGF−β1 and VEGF among all groups (Figure 3C).

The evaluation of the skin collected on day 7 post-lesion after staining with H&E showed that the animals of the group DMCAM had a lower number of leukocytes and a higher number of blood vessels in comparison to the CTSF and DMSF groups (Figure 4A, B). There was no difference in the analysis of fibroblasts and mastocytes. The Western blotting analysis for TGFβ1 and VEGF did not present statistical differences among the groups, in spite of the better expression of TGF−β1 in relation to the other groups (Figure 4C).

According to the qualitative and quantitative histopathological analysis of the skin collected on post-wound day 14 and stained with H&E, a smaller number of leukocytes and fibroblasts was observed in group DMCAM in comparison to groups CTSF and DMSF (*p* < 0,05). In group CTSF, a higher quantity of mastocytes was observed in relation to groups DMSF and DMCAM (Figure 5A,B). Western blotting analysis for TGF−β1 and VEGF did not demonstrate statistical differences (Figure 5C).

### 3.4. Distribution, Deposition, Organization, and Thickness of Collagen Fibers on the Skin Wounds of Hyperglycemic Animals Submitted to Topical Treatment with Plasticized Chitosan-Alginate Membranes

The qualitative histopathological analysis of the intact skin and of the samples collected on post-wound days 3, 7, and 14 were used to assess the distribution, deposition, organization, and thickness of the collagen fibers with the Picrosirius red technique. To visualize the elastic fibers, Weiger, Resorcin-Fuchsin was used. The elastic fibers are shown in black, and according to the increasing quantity of elastin associated with the microfibrils, they can be divided into three types: oxytalan, eulanin and mature.

There was no difference among groups on post-wound day 3 in regard to distribution, deposition, organization, and thickness of the collagen fibers (Figure 6A). On post-wound day 7, through both staining analyses, collagen fibers with a discreet red-orange birefringence were observed in the DMCAM group scars when compared to what could be observed in groups CTSF and DMSF (Figure 6A). As to the elastic fibers in the DMCAM group, a small concentration of thinner elastic fibers was observed, indicating the beginning of the production of this type of fiber at that time point (Figure 6B). On post-wound day 14, improved reorganization of the papillary dermis and enhanced red-orange birefringence of collagen fibers in the healing region of the DMCAM group was observed, as well as a network-shaped orientation, similar to intact skin (Figure 6A). As for the elastic fibers in the DMCAM group, there was an improvement in the reorganization of the elastic fibers, scattered on all of the neodermis, with a larger presence of eulanin microfibrils. This also shows a similar pattern to the elastic fibers of intact skin. Eulanin microfibrils are thicker and considered essential for the elastogenesis process of the skin (Figure 6B).

## 4. Discussion

Physiological inflammation is fundamental for tissue repair, however any imbalance in this phase can be harmful to the healing process. One of the consequences of a persistent inflammatory response in the wound bed is imbalance in macrophage proteolytic activity, which overwhelms the protective mechanisms of the tissue [26].

In practice, the use of dressings or the daily change of conventional bandages is an ongoing query. Hence, the purpose of this study was to assess the effect of using chitosan-alginate membranes plasticized with glycerol in the healing of excisional wounds, considering parameters such as angiogenesis, deposition of collagen matrix and elastic fibers on the skin wounds of hyperglycemic C57BL/6 mice, as well as relevant molecular markers in the healing process.

Previous studies demonstrated that dressing biomaterials made of chitosan and alginate improve the healing process of skin wounds because they offer protection against infection, provide a moist environment, avoid trauma, and can also release bioactive substances [12,21,27].

After skin injury, neutrophils and macrophages are attracted to the wound site. These cells release growth factors and pro-inflammatory cytokines that stimulate the formation of granulation tissue. When the recruitment of neutrophils and macrophages is disturbed, the formation of granulation tissue and wound closure are delayed [26]. In the inflammatory phase, three days post-wound, the chitosan-alginate membrane was able to modulate the number of mature leukocytes and mastocytes and increase the number of blood vessels. Balance during the inflammatory phase is crucial to repairing skin damage and decreasing the risks of local complications. Prolonged hyperglycemia can lead to a long inflammatory phase for skin wounds, hampering tissue repair. Investigating dressings that modulate the healing phase is important to decrease the healing time and consequently, decrease the risk of complications.

Another cell type capable of influencing healing is the mastocyte. A reduced number of these cells during the inflammatory phase can contribute to a decrease in inflammation and can therefore influence healing or the reduction of scar tissue [28].

The results observed during the histopathological analysis are summarized in Table 2.

During the proliferative phase, post-wound day 7, the resolution of the inflammatory phase could be observed, as well as the increase in the number of blood vessels with a better VEGF expression, despite no statistical difference being noticed. Besides, a discreet increase in the number of collagen fibers and elastic fibers was detected. Collagen is the most abundant protein found in the human body and one of the main components of the extracellular matrix. Collagen molecules are synthesized by fibroblasts, which are recruited to the wound area to proliferate and perform key activities with the purpose of modulating wound closure [29]. On the other hand, elastic fibers give the wound area elasticity and retraction, playing an essential role in the induction of biochemical responses and the reaction to mechanical forces derived from the microenvironment [30].

Results observed on post-wound days 3 and 7, regarding the modulation of the leukocytic infiltrate, are confirmed by a previous study in which it was observed that the use of chitosan-alginate membranes on the skin wounds of healthy animals could modulate the inflammatory phase, fibroplasia, and collagenesis, favoring the healing process [12]. In another study carried out with hyperglycemic animals and chitosan-alginate membranes, the modulation of IL−1α, IL−1β, and G-CSF pro-inflammatory cytokines (a granulocyte colony-stimulating factor) was observed, favoring tissue repair in a shorter period of time when compared to the control group [21]. We can observe that the glycerol added to the CAM ensures greater flexibility and does not alter the capacity to modulate tissue repair.

In the remodeling phase, on post-wound day 14, the inflammatory and proliferative phases are in remission, with a better resolution of the papillary dermis and reorganization of collagen and elastic fibers.

Tissue repair of wounds among people with diabetes is marked by high levels of reactive oxygen species (ROS), leading to oxidative stress and a prolonged inflammatory phase, contributing to a healing delay [4]. The presence of ROS above acceptable levels can lead to damage in different macromolecules found inside the cells, affecting the structure of nucleic acids and proteins, as well as the function of lipids of the cell membrane. Thus, these effects can be associated not only with inflammatory processes but also with early aging, development of tumors, and even neurodegenerative diseases [31]. Chitosan influences the redox state of cells and may contribute to prevent damage that results from reactions mediated by oxidative processes [31], protecting the cells. Furthermore, for several years, chitosan has been investigated as a scaffold component for the growth of cells in the fields of regenerative medicine and tissue engineering [32,33]. Therefore, it is widely indicated for dressings’ formulation.

When considering the ideal characteristics of a dressing for tissue repair of skin wounds, the following properties stand out: providing continuous protection, preventing proliferation of microorganisms, maintaining humidity balance, and favoring healing [34], as well as odor and pain reduction. The combination of chitosan and alginate is observed to successfully meet several of these requirements.

Chitosan promotes the formation of tissue granulation and collagen deposition with enhanced action on the dermo-epidermal wound [33], whilst alginate, being a very hydrophilic polymer, favors the balance between absorption and swelling of physiological fluids, which can be achieved in 12 h. Maintaining a moist environment can accelerate epithelial regeneration, contributing to healing wounds that do not close easily. In this sense, alginate can contribute positively since, when it absorbs the exudate, it forms a gel that fosters a moist environment, facilitating dressing changes, and reducing the pain and trauma associated with this process [35]. Both polymers are biocompatible and present hemostatic activity [36,37,38], and their combination in PEC form, besides keeping many of the features of both polymers individually, results in a transparent biomaterial that allows inspection of wound evolution without needing to remove the dressing. The addition of glycerol to the formulation further improved the easiness of application and the mechanical properties of the membrane.

Furthermore, another important component of the conjunctive tissue are the elastic fibers of the extracellular matrix, composed of elastin and fibrils that are responsible for skin elasticity [39]. Normally these fibers have a long half-life and do not regenerate post-wound. However, results obtained in this study demonstrate that applying the plasticized chitosan-alginate membrane improved elastic fiber deposition. Healing of skin wounds is a dynamic process that demands the participation of numerous cells and molecules. Dressings produced with products of biological origin have been in the spotlight for the last decades since they present biological activity and even structural components that mimic the extracellular matrix, hence promoting cell attraction, adherence, and proliferation. Therefore, they demonstrate great potential for use in wounds that do not heal easily, and future investigation of cellular markers associated with plasticized chitosan-alginate membranes needs to be addressed.

## 5. Conclusions

The use of plasticized chitosan-alginate membranes in the skin wounds of hyperglycemic animals promoted tissue repair through a decreased number of inflammatory cells and mastocytes, improving collagen and collagen fibers deposition. Hence, it demonstrates a high potential for skin wound dressings in hyperglycemic patients.

## Figures and Tables

**Figure 1 polymers-14-04754-f001:**
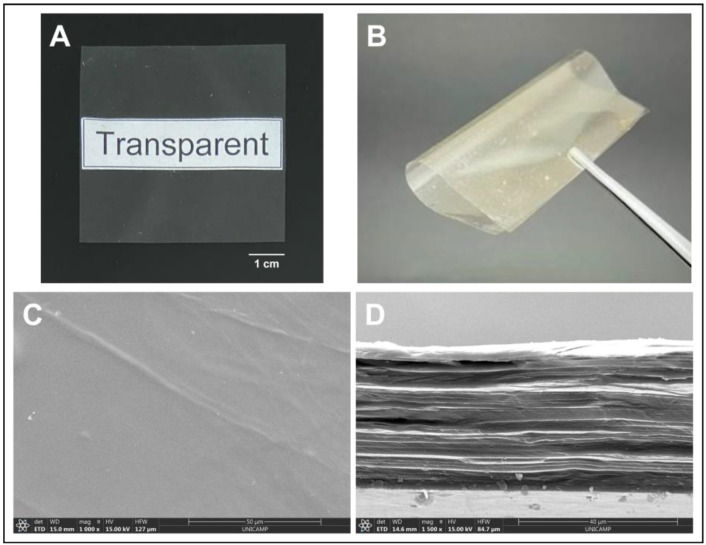
Typical visual aspect of the membrane produced. Images showing (**A**) the transparency and (**B**) flexibility of the membrane analyzed with the naked eye; images showing the membrane (**C**) microstructure of the surface and (**D**) cross-section of the membrane by SEM.

**Figure 2 polymers-14-04754-f002:**
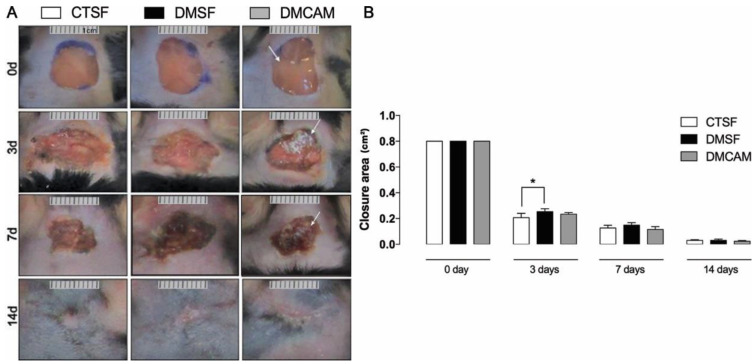
(**A**) representative images of the healing process were monitored at day 0, 3, 7, 14 after injury. (**B**) Wound areas were quantified and expressed as closure area (cm^2^), and area analyzed with ImageJ^®^ of normoglycemic animal groups submitted to topical treatment with 0.9% saline solution (CTSF group); of hyperglycemic animals submitted to topical treatment with 0.9% saline solution (DMSF group); and hyperglycemic animals submitted to a topical solution with chitosan-alginate membranes (DMCAM group) on post-wound days 3, 7 and 14. Arrows indicate the presence of alginate and chitosan membranes (CAM) in the wound bed. * Groups statistically different (*p* < 0.05), compared by One-way ANOVA, followed by a Tukey multiple comparison test.

**Figure 3 polymers-14-04754-f003:**
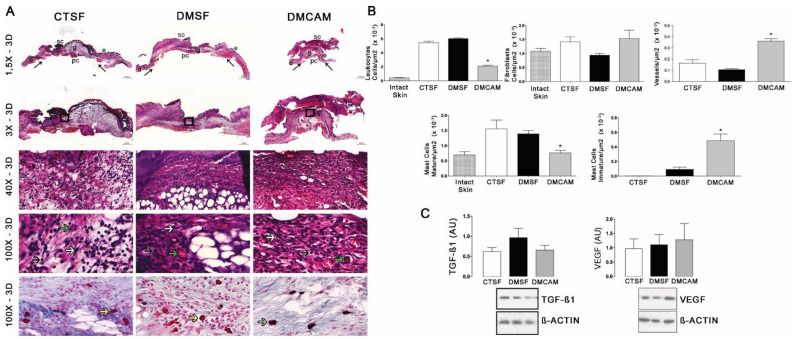
Topical treatment with plasticized chitosan-alginate membranes in hyperglycemic mice reduced the number of inflammatory cells and fibroblasts and increased the density of new vessels in the wound bed. (**A**) photos representing skin stained with H&E, post-wound day 3; less enhanced black arrows indicate the location where the excisional wound was performed. Small letters represent: sc, scab; e, epidermis; g, granulation tissue; d, dermis; pc, panniculus carnosus. Boxes in the upper panels indicate areas where regions that were amplified are shown to illustrate leukocytes (black arrows), fibroblasts (white arrows), and new vessels (green arrows). (**B**) number of leukocytes, fibroblasts, vessels, and blood capillaries by μm^2^ × 10^−3^; at day 3, the DMCAM group showed significantly smaller number of leukocytes and mature mastocytes than CTSF and DMSF groups. The DMCAM group showed significantly increased numbers of vessels, blood capillaries, and also a higher number of active mastocytes than CTSF and DMSF. Histological stain with Alcian-Blue + Safranin on skin fragments on day 3 after creating the wound on the backs of mice for analysis of mature mastocytes (yellow arrows) and immature mastocytes (grey arrows). (**C**) Analysis of the skin fragment (wound) with Western Blotting, with VEGF and TGF−β1 and Beta Actin (Control), of animal groups CTSF, DMSF and DMCAM, on post-wound day 3. * Groups statistically different (*p* < 0.05), compared by One-way ANOVA, followed by a Tukey multiple comparison test. Enlargements of 1.5×, 3×, 40× e 100×; scale bar = 1000 μm, 500 μm, 50 μm and 20 μm.

**Figure 4 polymers-14-04754-f004:**
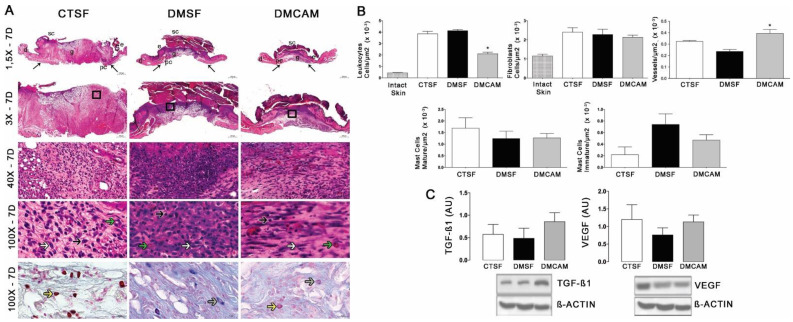
Topical treatment with plasticized chitosan-alginate membranes in diabetic mice reduced the number of inflammatory cells and fibroblasts and increased the density of new vessels in the wound bed. (**A**) photos representing skin stained with H&E on day 7 of the wound. Less enhanced black arrows indicate the location where the excisional wound was performed. Small letters represent: sc, scab; e, epidermis; g, granulation tissue; d, dermis; pc, panniculus carnosus. Boxes in the upper panels indicate areas where regions that were amplified are shown to illustrate leukocytes (black arrows), fibroblasts (white arrows), and new vessels (green arrows). (**B**) number of leukocytes, fibroblasts, vessels, and blood capillaries by μm^2^ × 10^−3^; at day 7, the DMCAM group showed a significantly lower number of leukocytes in comparison to the CTSF and DMSF; a number of blood vessels significantly higher was observer in the DMCAM group in comparison than to those in the CTSF and DMSF groups. Histological stain with Alcian-Blue + Safranin on skin fragments on day 7 after creating the wound on the backs of mice for analysis of mature mastocytes (yellow arrows) and immature mastocytes (grey arrows). (**C**) Analysis of the skin fragment (wound) with Western Blotting, with VEGF and TGF−β1 and Beta Actin (Control), of animal groups CTSF, DMSF, and DMCAM, on post-wound day 7. * Groups statistically different (*p* < 0.05), compared by One-way ANOVA, followed by a Tukey multiple comparison test Enlargements of 1.5×, 3×, 40× e 100×; scale bar = 1000 μm, 500 μm, 50 μm and 20 μm.

**Figure 5 polymers-14-04754-f005:**
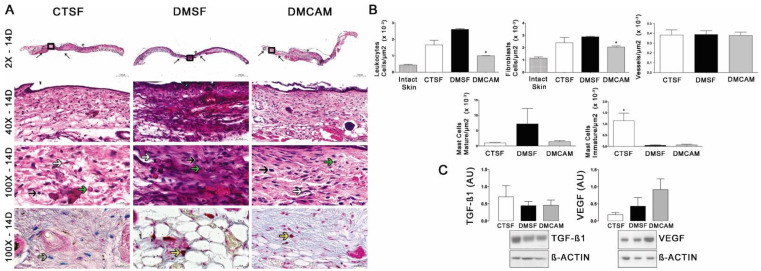
Topical treatment with plasticized chitosan-alginate membranes in diabetic mice reduced the number of inflammatory cells and fibroblasts in the wound bed. (**A**) photos representing skin stained with H&E on day 14 of the wound. Less enhanced black arrows indicate the location where the excisional wound was performed. Small letters represent: sc, scab; e, epidermis; g, granulation tissue; d, dermis; pc, panniculus carnosus. Boxes in the upper panels indicate areas where regions that were amplified are shown to illustrate leukocytes (black arrows), fibroblasts (white arrows), and new vessels (green arrows). (**B**) number of leukocytes, fibroblasts, vessels, and blood capillaries by μm^2^ × 10^−3^; at day 14, the DMCAM group showed significantly lower numbers of leukocytes and fibroblasts in comparison to the CTSF and DMSF groups; the CTSF group showed a significantly higher quantity of mastocytes than the DMSF and DMCAM groups. Histological staining with Alcian-Blue + Safranin on skin fragments on day 14 after creating the wound on backs of mice for analysis of mature mastocytes (yellow arrows) and immature mastocytes (grey arrows). (**C**) Analysis of the skin fragment (wound) with Western Blotting, with VEGF and TGF−β1 and Beta Actin (Control), of animal groups CTSF, DMSF, and DMCAM, on post-wound day 14. * Groups statistically different (*p* < 0.05), compared by One-way ANOVA, followed by a Tukey multiple comparison test. Enlargements of 1.5×, 3×, 40× e 100×; scale bar = 1000 μm, 500 μm, 50 μm and 20 μm.

**Figure 6 polymers-14-04754-f006:**
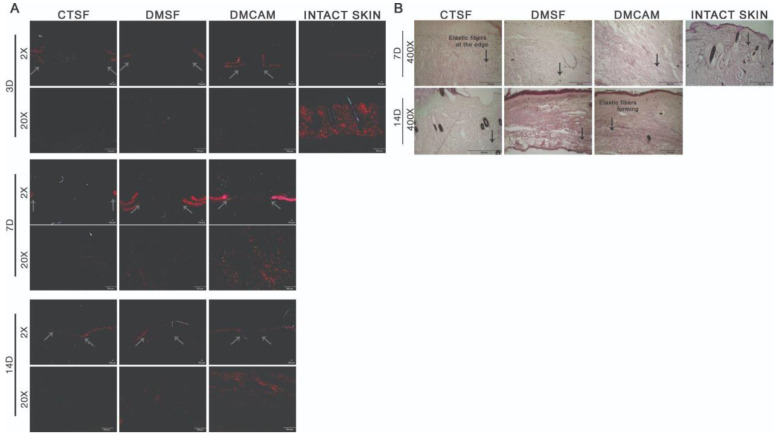
Reorganization of collagen fibers in neodermis after topical application with plasticized chitosan-alginate membranes and deposition of elastic fibers after the excisional wound. (**A**) skin fragments with wounds collected on days 3, 7, and 14 showing images representing birefringence of collagen fibers stained with Picrosirius red and analyzed under polarized light. Less enhanced grey arrows indicate the location where excisional wounds were performed. Enlargements of 2× and 20×, scale bar = 100 μm. All groups were compared with intact skin fragments stained with Picrosirius red (right side box). (**B**) Skin wound fragments collected after days 7 and 14 with images representing collagen fibers stained with the Weigert Resorsina-Fucsina method to identify elastic fibers. Less enhanced black arrows indicate the aspect of stained elastic fiber. Enlargements of 400×; scale bar = 100 μm. All groups were compared to intact skin fragments (G) stained with Weigert Resorsina-Fucsina (right side box).

**Table 1 polymers-14-04754-t001:** Properties of the CA membranes plasticized with glycerol used in the in vivo experiments.

Property		Value (Average ± SD)
Thickness	Dry (mm)	0.056 ± 0.013
	After exposure to saline solution at 37 °C for 24 h (mm)	0.311 ± 0.009
Opacity (%)		0.0 ± 0.0
Contact angle with water (°)		45 ± 6
Swelling in saline solution (after 24 h at 37 °C) (g/g)		7.5 ± 0.7
Mass loss in saline solution (after 7 days at 37 °C) (%)		16.0 ± 1.7
Average rugosity	Face exposed to air during drying (µm)	2.09 ± 0.53
	Face exposed to Petri dish surface during drying (µm)	1.29 ± 0.43
Tensile strength (MPa)		85.4 ± 24.0
Elongation at break (%)		5.1 ± 2.5
Young’s module (MPa)		4024 ± 477

**Table 2 polymers-14-04754-t002:** Histomorphological evaluation performed on days 3, 7 and 14, in which the occurrence intensity of leukocytes, fibroblasts, blood vessels, mature and immature mastocytes was analyzed. The following scoring system was used in the evaluation: − for absent; + for light; ++ for moderate; and +++ for iIntense responses. The assessment consisted of a slide scan and comparisons of the groups.

Group	Time of Analysis (Days)	Intensity of Response Level				
		Leukocytes	Fibroblasts	Blood Vessels	Mature Mastocytes	Immature Mastocytes
CTSF	3	+++	+++	+	+++	−
	7	+++	+++	++	+++	+
	14	++	+++	+++	+	+++
DMSF	3	+++	+++	+	+++	+
	7	+++	+++	+	++	+++
	14	+++	+++	+++	++	+
DMCAM	3	+	+++	+++	+	+++
	7	+	+++	+++	++	++
	14	+	++	+++	+	+

## Data Availability

All data that our analyses relied on will be shared in an open access database (https://redu.unicamp.br/, accessed on 1 November 2022) as soon as the manuscript is approved.

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
