# Peer review of "Histological and Molecular Evidence of the Positive Performance of Glycerol-Plasticized Chitosan-Alginate Membranes on Skin Lesions of Hyperglycemic Mice"

_polymers, 2022, doi:10.3390/polym14214754_

Round 1

Reviewer 1 Report

The authors submitted a manuscript entitled “Histological and molecular evidence of the positive performance of glycerol-plasticized chitosan-alginate membranes on skin lesions of hiperglycemic mice” with the reference polymers-1932218.

The subject of the manuscript is quite important not only in the materials field but also on what concerns medicinal and histological areas (with an important stress on the diabetes).

The article is very well designed, the English is very good (almost without any flaws), all the methodology used, the design of the experiments, the ethical care, the framing of the importance that this type of studies can have in the area of diabetes, all they feature high quality. Overall, the manuscript is relevant, the results sound, the references well-presented and adequate to the Polymers magazine.

The only suggestions to improve the manuscript are in the graphical part:

In Figure 1A to better show the transparency of the membrane maybe a picture of an object or text behind the membrane it would be a better choice.

Figure 2 will be clearer if the right part could be a little bit enlarged. If this is difficult horizontally, it is suggested to widen a little vertically and also increase the size of the letters.

Figures 3, 4 and 5 it is suggested that the font size of parts B and C is also increased. In these figures, and according to the captions, the 100X-3D bottom row images belong to part B of the figures. This is the only way to justify referring to the red and gray arrows in the captions in this part of the captions. As such the images must be changed so that the B is higher up.

Apart from these minor graphic corrections, the manuscript is ready to be published.

Author Response

Thank you very much for the questions and suggestions, which showed that our work was thoroughly analysed. We appreciate all your comments and recommendations. The respective responses are given below, in blue, and the corresponding alterations in the manuscript  are highlighted in yellow.

We hope that the present version of the manuscript reaches more closely the high standard requirements of the Polymers journal. Feel free to contact us in case any further clarifications are needed.

Thank you again for the recommendations that certainly improved the overall quality of our work.

Reviewer 1

In Figure 1A to better show the transparency of the membrane maybe a picture of an object or text behind the membrane it would be a better choice.

Thank you for the suggestion. A new picture was taken, as recommended, with a reference in the background (the word “Transparent”), and is substituting image A in Figure 1 in the present version of the manuscript. We changed all letters of identification of  the images in Figure 1 to capital letters to maintain the same style of the other parts of the manuscript.

Figure 2 will be clearer if the right part could be a little bit enlarged. If this is difficult horizontally, it is suggested to widen a little vertically and also increase the size of the letters.

Thank you for the suggestion, we widened a little vertically and also increased the size of the letters, as recommended.

Figures 3, 4 and 5 it is suggested that the font size of parts B and C is also increased. In these figures, and according to the captions, the 100X-3D bottom row images belong to part B of the figures. This is the only way to justify referring to the red and grey arrows in the captions in this part of the captions. As such the images must be changed so that the B is higher up.

 Thank you for the suggestion, we enlarged the font sizes of images B and C in the revised version of Figures 3, 4, and 5. 

Reviewer 2 Report

The manuscript of Azevedo et al. describes tissue repair of excisional wounds in hyperglycemic animals treated with chitosan-alginate membranes (CAM) produced in the presence of glycerol. The animals were grouped into normoglycemic animals with a 0.9% saline solution topical treatment (CTSF); hyperglycemic animals with 0.9% saline solution topical treatment (DMSF) and hyperglycemic animals with glycerol-plasticized chitosan-alginate membrane topical treatment (DMCAM). The authors claimed that the DMCAM group presented a lower number of leukocytes, mature mastocytes, a higher number of vessels as compared to other treatments.

Major points:

1) Fig. 1: please add scale bars.

2) Fig. 2: The legend should be modified and precise. It should be a) in vivo images, b) quantification. The diagram is too small; it is impossible to read it. Please add scale bars to a).

3) Fig. 3: a: scale bars are missing.

Diagrams are too small.

What is the significance (p values) in C?

4) Fig. 4: a: scale bars are missing.Diagrams are too small.

What is the significance (p values) in B &C?

5) Fig. 5: a: scale bars are missing.

Diagrams are too small.

What is the significance (p values) in C?

6) Fig. 6: a: scale bars are too small, not readable at all.

A-B: The quality of the images is very low. It is impossible to draw any conclusions from the images.

I do not see properly any collagen fibers and any elastin fibers.

7) Very poorly written discussion, I do not understand the point of writing about inflammation, if there is no inflammation data shown:

In the inflammatory phase, three days post-wound, the chitosan-alginate membrane was able to modulate the number of mature leukocytes and mastocytes and increase the number of blood vessels. Balance during the inflammatory phase is crucial to repairing skin damage and decreasing the risks of local complications. Prolonged hyperglycemia can lead to a long inflammatory phase for skin wounds, hampering tissue repair. Investigating dressings that modulate the healing phase is important to decrease the healing time and consequently, decrease the risk of complications. Another cell type capable of influencing healing is the mastocyte. A reduced number of these cells during the inflammatory phase can contribute to a decrease in inflammation and can therefore influence healing or the reduction of scar tissue(28).

It is a pure speculation that DMCAM has an anti-inflammatory effect because it is not shown in the paper.

Author Response

Thank you very much for the questions and suggestions, which showed that our work was thoroughly analysed. We appreciate all your comments and recommendations. The respective responses are given below, in blue, and the corresponding alterations in the manuscript  are highlighted in yellow.

We hope that the present version of the manuscript reaches more closely the high standard requirements of the Polymers journal. Feel free to contact us in case any further clarifications are needed.

Thank you again for the recommendations that certainly improved the overall quality of our work.

Reviewer 2

1)  Fig. 1: please add scale bars.

The scale bars were added in the lower right side, thank you for pointing out this issue. 

2) Fig. 2: The legend should be modified and precise. It should be a) in vivo images, b) quantification. The diagram is too small; it is impossible to read it. Please add scale bars to a).

Thank you for the suggestion. We enlarged the diagram,  added the scale bars (but they are quite small, given that the images themselves are little) and identified the different parts of Figure 2 as A and B,  as recommended. We also improved Figure 2 caption as indicated in yellow in the manuscript.

3) Fig. 3: a: scale bars are missing. Diagrams are too small. What is the significance (p values) in C?

Thank you for the comments. We enlarged the diagram, and scale bars were added in the lower right side ((but they are quite small, given that the images themselves are small). We also commented on the statistical significance in the figure caption, as shown in yellow in the manuscript.

4) Fig. 4: a: scale bars are missing. Diagrams are too small. What is the significance (p values) in B?

Thank you for the suggestions. We enlarged the diagrams, and scale bars were added in the lower right side (but they are quite small, given that the images themselves are little). We added information of statistical significance and changed Figure's 4 caption  in the manuscript (shown in yellow).

5) Fig. 5: a: scale bars are missing. Diagrams are too small. What is the significance (p values) in B?

Thank you for the suggestions. We enlarged the diagrams, and scale bars were added in the lower right side (but they are quite small, given that the images themselves are little). We added information of statistical significance and changed Figure's 5 caption  in the manuscript (shown in yellow).

6) Fig. 6: a: scale bars are too small, not readable at all. A-B: The quality of the images is very low. It is impossible to draw any conclusions from the images. I do not see properly any collagen fibers and any elastin fibers.

To improve the visualisation of the results, we enlarged the diagram, but the scale bars  are quite small, given that the images themselves are little. We also increased the size of the imagens and added the black arrows for better identification in image 6B.

7) Very poorly written discussion, I do not understand the point of writing about inflammation, if there is no inflammation data shown. It is a pure speculation that DMCAM has an anti-inflammatory effect because it is not shown in the paper.

Some of the responses analysed are directly related to inflammation, as described below and included in the present version of the manuscript, for better contextualization. We thank the reviewer for pointing out this issue.

“After skin injury, neutrophils and macrophages are attracted to the wound site. These cells release growth factors and pro-inflammatory cytokines that stimulate the formation of granulation tissue. When the recruitment of neutrophils and macrophages is disturbed, the formation of granulation tissue and wound closure are delayed (26)”

Reviewer 3 Report

Dear Authors,

I consider your paper has to be improved before being accepted for being published, in order to highlight the element of novelty introduced by your research. 

Author Response

Thank you very much for the questions and suggestions, which showed that our work was thoroughly analysed. We appreciate all your comments and recommendations. The respective responses are given below, in blue, and the corresponding alterations in the manuscript  are highlighted in yellow.

We hope that the present version of the manuscript reaches more closely the high standard requirements of the Polymers journal. Feel free to contact us in case any further clarifications are needed.

Thank you again for the recommendations that certainly improved the overall quality of our work.

Reviewer 3

I consider your paper has to be improved before being accepted for being published, in order to highlight the element of novelty introduced by your research. 

Thank you for the comment. We had highlighted the novelty of the work in the last paragraph of the Introduction, and included an additional phrase in its end to stress it in the revised version of the manuscript.

Reviewer 4 Report

Introduction would benefit from a small paragraph explaining what to be expected in the entire manuscript.

Can you mention please the city and country of the Laboratories were the mice were obtained from?

Would you clarify please the reason to inject 0.1 M citrate buffer for 5 days in Section 2.2?

Section 2.5: Do you want to mention that you checked/assessed, or you already “observed” those animal behaviours. I would mention “checked” or “assessed” as it gives the impression that all the animals were with those signs during the experiments.

Section 2.7: should be “the final membranes were then characterized regarding their aspect…”

Angstrom unit and all the rest of units should be corrected as sometimes they’re missing either subscripts or superscripts.

Section 2.11: pH 7.4 is repeated and is confusing the whole sentence. I suggest removing the first one, so you can keep the flow of the sentence.

Sec 2.11: Please correct to “…at room temperature for 2 h with slow shaking.” As well, the next paragraph has the same mistake “shaken slightly”, it should be “slow shaking”. “Two h” should be “2 h”

Table 1. the number of decimals after coma should be kept the same in all cases.

Fig. 1: The caption text is quite confusing. There is not text addressed to Fig 1b. As well, I am convinced that you show how clear is the membrane in Fig. 1a. I would leave Fig 1b, which clearly shows that the membrane is transparent and flexible.

Fig 2. Needs a scale bar. Please add the scale bars.

Do you want to stick to HE or H&E abbreviation throughout the manuscript?

Fig. 3 and 4. Red colour arrows should be replaced with another colour as is quite hard to distinguish the difference between mastocytes or the arrows.

To summarise the efficiency of chitosan-alginate membranes in vivo for the H&E staining, the authors should make a scoring list/table to show.

When mentioning about chitosan applicability and uses, please add some supporting references.

Author Response

Thank you very much for the questions and suggestions, which showed that our work was thoroughly analysed. We appreciate all your comments and recommendations. The respective responses are given below, in blue, and the corresponding alterations in the manuscript  are highlighted in yellow.

We hope that the present version of the manuscript reaches more closely the high standard requirements of the Polymers journal. Feel free to contact us in case any further clarifications are needed.

Thank you again for the recommendations that certainly improved the overall quality of our work.

Reviewer 4 

1) Introduction would benefit from a small paragraph explaining what to be expected in the entire manuscript.

We added a sentence at the end of the Introduction section addressing this issue.

2) Can you mention please the city and country of the Laboratories were the mice were obtained from? 

We added the following information in the revised version of the manuscript: Campinas, SP, Brazil.

3) Would you clarify please the reason to inject 0.1 M citrate buffer for 5 days in Section 2.2? 

0.1 M citrate buffer was the solvent used to suspend  the streptozotocin injected in the animals for five consecutive days to induce diabetes. With the procedure of injecting only the citrate buffer we aimed to submit all animals to a similar stress condition independently if they were or not exposed to the diabetes induction procedure.

Section 2.5: Do you want to mention that you checked/assessed, or you already “observed” those animal behaviours. I would mention “checked” or “assessed” as it gives the impression that all the animals were with those signs during the experiments. 

Thank you for the observation. As suggested, “observed” was substituted by “assessed” to better describe what was actually done. A change in the sentence was done to improve clarity and is show in yellow in the present version of the manuscript.

4) Section 2.7: should be “the final membranes were then characterized regarding their aspect…” Angstrom unit and all the rest of units should be corrected as sometimes they’re missing either subscripts or superscripts. 

Thank you for noticing these. The term “their” was included in the sentence and all units subscripts/superscripts were revised.

5) Section 2.11: pH 7.4 is repeated and is confusing the whole sentence. I suggest removing the first one, so you can keep the flow of the sentence.

Thank you for the suggestion. We deleted the term.

6)  Sec 2.11: Please correct to “…at room temperature for 2 h with slow shaking.” As well, the next paragraph has the same mistake “shaken slightly”, it should be “slow shaking”. “Two h” should be “2 h” Table 1. the number of decimals after coma should be kept the same in all cases. 

We changed “shaken slightly” in both sentences  to “under mild shaking”. We also have done the other suggestions recommended.

7) Fig. 1: The caption text is quite confusing. There is not text addressed to Fig 1b. As well, I am convinced that you show how clear is the membrane in Fig. 1a. I would leave Fig 1b, which clearly shows that the membrane is transparent and flexible. 

Figure 1A was retaken to better show the transparency of the membrane. We kept both images so that the reader can better understand and visualise the type of biomaterial we have used in this study.

8) Fig 2. Needs a scale bar. Please add the scale bars. 

The modifications were performed as requested,however the scale bars  are quite small, given that the images themselves are little.

9) Do you want to stick to HE or H&E abbreviation throughout the manuscript? 

We standardised the term throughout the text to H&E.

10) Fig. 3 and 4. Red colour arrows should be replaced with another colour as is quite hard to distinguish the difference between mastocytes or the arrows. 

The colour of the arrows were changed, thank you for the suggestion.

11) To summarise the efficiency of chitosan-alginate membranes in vivo for the H&E staining, the authors should make a scoring list/table to show. 

The scoring table was introduced in the text. Thank you for the recoomendation.

12) When mentioning about chitosan applicability and uses, please add some supporting references.

Thank you for your comment. To better support the use of  chitosan, two references were added in section 4, as shown in yellow in the present version of the manuscript. On section 1 no references were additionally included since the information presented about chitosan was already based on 8 references.

Round 2

Reviewer 2 Report

The quality of the manusript has been improved.